# Soybean Yield and Nutrition Grown on the Straw of Grain Sorghum Inoculated with *Azospirillum brasilense* and Intercropped with BRS Paiaguás Grass

**DOI:** 10.3390/plants12102007

**Published:** 2023-05-17

**Authors:** Deyvison de Asevedo Soares, Viviane Cristina Modesto, Allan Hisashi Nakao, Wellington Rosa Soares, Leandro Alves Freitas, Lourdes Dickmann, Isabô Melina Pascoaloto, Marcelo Andreotti

**Affiliations:** 1Department of Soil Science, University of Sao Paulo—College of Agriculture “Luiz de Queiroz”, Piracicaba 13418-900, Brazil; 2Department of Plant Health, Rural Engineering and Soils, Faculty of Engineering of Ilha Solteira/UNESP, Ilha Solteira 15385-000, Brazil

**Keywords:** *Glycine max* L., *Sorghum bicolor* L. Moench, intercropping, grain yield, soil fertility

## Abstract

The adoption of diversified agricultural systems that employ integrated cultural practices appears to be the way to sustainably intensify tropical agriculture. Our objectives were to evaluate the dry matter (DM) accumulation of sorghum inoculated with *Azospirillum brasilense*, with or without a nitrogen fertilization split, intercropped with palisade grass (*Urochloa brizantha* cv. BRS Paiaguás), and how these practices influenced the nutrition and development of soybean in succession. The design was a randomized complete block in a 2 × 2 × 3 factorial, consisting of sorghum monoculture cropped or intercropped with palisade grass, sorghum either inoculated or not with *A. brasilense*, and nitrogen applied at 120 kg ha^−1^ N only at sowing, only at topdressing, or split—30% at sowing and 70% at topdressing at the beginning of the panicle initiation stage. The residual impacts of these treatments on the following soybean crop were also evaluated. Higher DM yield occurred in sorghum inoculated with *A. brasilense*, however, this result varied by year. The sorghum–palisade grass intercrop produced a higher amount of straw than sorghum monoculture. The nutrition of soybean was adequate regardless of treatments, but grain yield was higher when the sorghum residue was inoculated. The inoculation of *A. brasilense* in sorghum intercropped with palisade grass increased yield. The nutrition of soybean was adequate regardless of the treatments, while grain yield was higher on the inoculated sorghum residues. The inoculation of *A. brasilense* in sorghum intercropped with palisade grass increased DM yield. The intercropping increased the production of biomass for animal grazing and DM for soil coverage. The inoculation of sorghum by *A. brasilense* and its intercropping with palisade grass contributed to higher soybean yield in succession.

## 1. Introduction

Recently, the world population reached 8 billion people and is expected to reach 9 billion by 2037 [1]. With global population growth, the demand for food is increasing, but due to limited availability of new agricultural land, agriculture has been intensified [2].

Emerging low-income regions, with greater land availability, are expected to experience up to 50% of the global growth in agricultural production this decade, mainly in Sub-Saharan Africa and Latin America (17 and 15%, respectively), where Brazil and Argentina are expected to intensify the use of already cultivated lands [3]. In this scenario, the dependence on fertilizers for the intensification of agriculture makes the development of more sustainable agricultural practices indispensable [2].

The Brazilian cerrado has witnessed a great advance in conservation production systems, associating the no-tillage system (NT) with integrated crop–livestock systems (ICLS). The diffusion of these technologies has been important in tropical regions, due to the predominance of soils with low natural fertility [4] and climatic conditions generally limiting for a second crop in the fall/winter in these regions [5,6,7]. In view of this, practices for the improvement of soil fertility, allied to the development of cultivars adapted to such conditions, have subsidized the increase in food productivity in tropical agriculture in recent decades [8,9].

The success and consolidation of NT are mainly due to the use of plants adapted to each condition [10]. Species that produce enough straw to cover the surface of the soil, with a slow degradation rate, should be chosen, but the production and amount of nutrients accumulated in the straw also depend on the management of fertilization, phenological stage of desiccation, C/N relationships and lignin/N, type of soil, and climatic conditions [11], among others. New technologies, such as biological inoculants for plant growth promotion, have also been adopted as part of an integrated management; however, it is necessary to evaluate the interactions between these biological inputs and factors such as fertilizations and consortia, as well as the effects of these practices on successive crops.

The integration of agronomic practices associated with the intensification of intercropping systems, tropical forages grasses of the genus *Urochloa* (Syn. *Brachiaria*), and grain crops, for the production of grains in the off-season and forage and/or straw for the summer season, has been widely disseminated in the Brazilian cerrado as components of ICLS [12,13,14]. In this modality, corn has been chosen as the main crop, but other grasses have been evaluated as an alternative, such as the sorghum crop (*Sorghum bicolor* L. Moench).

Grain sorghum is widely used as a substitute component to corn in animal feed formulations, being an excellent energy resource, and it can also be a human food component (for example, breads and cakes). The crop has proven to be an excellent alternative for grain and forage production in intercropping with palisade grass (*Urochloa brizantha*) in the ICLS, due to its greater resilience and high biomass yield under edaphoclimatic conditions, which are limiting for other grasses [15,16,17,18]. The intercropping of sorghum with palisade grass, therefore, seems to be a promising system for the conditions of the Brazilian cerrado, since in addition to the remaining biomass of sorghum after grain harvesting, the palisade grass has a deep and voluminous root system [19,20] and produces large amounts of biomass for grazing in the off-season and/or soil cover for the crop in succession [21,22], usually soybean.

Soybean [*Glycine max* L. (Merr.)] is one of the most important crops worldwide [23], with Brazil, USA, and Argentina being the major producers; 38%, 31%, and 13% of global production in 2021, respectively [24]. In Brazil, soybean is the most planted field crop [25] and the most important crop that contributes to total crop cash receipts [26], but the average crop yield is still considered low (3.527 kg ha^−1^) [27]. In addition, studies are necessary for developing practices that improve the crop yield while at the same time maintaining or improving environment sustainability.

The path to intensifying tropical agriculture and the sustainability of agricultural systems appears, therefore, to come from the adoption of practices that result in more complex systems, where cover crops cycle considerable amounts of nutrients from well-developed root systems, which intercept nutrients from deep layers and release them on the soil surface [28,29] to benefit the development of crops in succession [30,31].

In this sense, the hypotheses of this study are: inoculated grain sorghum with *Azospirillum brasilense* produces more biomass; the partitioning of nitrogen (N) fertilization influences the inoculation of *A. brasilense* in sorghum; and the effects of *A. brasilense* and the partitioning of nitrogen fertilization in sorghum influence the soybeans’ development and yield in succession. Our objectives were to evaluate the accumulation of the dry matter of grain sorghum inoculated with *A. brasilense*, with or without the partitioning of nitrogen fertilization, and intercropped with palisade grass, and how these practices influenced the nutrition and development of soybeans grown in succession.

## 2. Results and Discussion

Significant interactions among the evaluated factors were not observed (*p* ≥ 0.05) for the accumulation of dry matter of the aerial part (DMAP) of sorghum. However, in 2015 there was an isolated effect for the accumulation of DMAP, which positively influenced the intercropping with inoculated sorghum (Figure 1). In this case, it is possible that the bacteria migrated to the grass, so that both species benefited from the growth-promoting effects conferred by genus *Azospirillum* [32,33,34], since both can host this genus of bacteria [35]. Under the same conditions as in the present study, Nakao et al. [36] reported that inoculation of grain sorghum seeds with *A. brasilense* increased the dry matter yield of the aerial part of both monoculture sorghum and intercropped with grass.

Nitrogen fertilization management did not influence the accumulation of DMAP (Figure 1). Although fertilization was carried out at different timings and in varying split applications, the total amount applied was the same in all treatments (120 kg N ha^−1^). Additionally, the clay texture of the soil contributed to lower N losses from mineral fertilizers [37], and the mineralization of OM itself may have contributed to the availability of N in the soil solution [38,39] throughout the intercropping cycle, in moments of N deficit in the splitting intervals, since the OM content in the soil was appreciable (23.0 g dm^−3^). This hypothesis should be tested in soils with low natural fertility, such as those predominant in tropical agricultural frontiers.

Nitrogen dynamics in the soil–plant atmosphere system are complex and their efficiency depends on several aspects interacting simultaneously; cropping system, amount and time of N application, plant residues from previous harvests, and edaphoclimatic conditions that act in the volatilization, leaching, and N mobilization or immobilization [40,41,42,43]. However, NT history in the studied area may have provided the necessary stability so that the availability of N would not be a limiting factor for the development of intercropped crops.

Although studies have shown that intercropping forage grasses with grain crops can reduce DMAP yield in the main crop [36,44,45], the literature presents divergent results, since this behavior depends on edaphic factors, climatic conditions, and the growth habit of the intercropped species, causing normal development of the main crop in the intercrop, or even superior to its exclusive cropping. Regardless of such divergences, our results indicated that, for biomass yield purposes and NT maintenance, the amount of biomass produced offsets due to the higher DM input into the system compared to the monoculture.

Regarding soybean nutrition over the two years, there was no significant interaction among any of the evaluated factors (Table 1). In the 2015/16 season, foliar levels of N, K, and Ca in soybean on residues without inoculation were higher (*p* < 0.05). In 2016/17, in addition to these nutrients, the P content was also higher in the treatment without inoculation. Except for K, in the second year, the other macronutrients remained within adequate levels for soybean nutrition [46].

Leaf K content was below the minimum adequate limit for soybean nutrition in all treatments in the 2015/16 season. According to Malavolta [47], more than 80% of the K present in plant tissue is in soluble form; this is due to the fact that the nutrient does not have a structural function in the plant tissue [48], which causes its loss by relatively rapid leaching into the soil. In addition, Rosolem et al. [49] reported that, on medium-textured soil, a rainfall of only 8.7 mm caused high K release from sorghum and *Urochloa* straw; therefore, although the present study was carried out in a soil with a clayey texture (580 g kg^−1^), a hypothesis for the low foliar levels of K in soybean would be the asynchrony between the release of nutrients from the residues and the soybean demand, since the precipitation was concentrated and had irregular distribution in time (Figure 2).

Regarding the sowing methods, soybean grown on intercropping residues in 2015/16 showed a higher leaf concentration of chlorophyll (SPAD index), Ca, and Mg (Table 1). This result can be explained by the greater amount of biomass produced in the intercrop (sorghum + *Urochloa*), which resulted in greater amounts of waste with proportional nutrients supplied to the crop in succession [11,45,50].

In 2016/17, the soybeans on the intercropped sorghum residues also had the highest leaf Mg content; on the other hand, the soybeans on sorghum monoculture residues showed the highest levels of N, P, and K (Table 1). These results in the last year differ from those found by Pariz et al. [50]; they reported that the greater amount of straw produced in the intercropping grain crops with grasses (forage and corn) increased the amounts of N, P, and K on the soil surface. In the present study, the better soybean nutrition on sorghum monoculture residues may have been caused by the lower interspecific competition in this treatment, due to lower plant density; however, this result did not influence the grain yield (GY), probably due to the nutritional levels being adequate [46] regardless of differences.

For the fertilization factor, there was no effect of N management in the previous intercropping on the nutrition of soybean plants in succession. These results corroborate the soybean response regarding the production components evaluated in terms of fertilization management, which also did not differ significantly. This can be justified due to the history of NT in the studied area, which provided stability in the availability of these nutrients for soybean nutrition.

In 2015/16, the lowest final plant stand (FPS) occurred on the residual of the inoculated intercropping (Table 2A); in this treatment, the largest amounts of sorghum and palisade grass residues were contributed (Figure 1). This result may have been associated with the possible negative interference of the straw at the time of sowing, when “lodging” eventually occurred due to the withered condition of the straw. This effect underscores the importance of considering the condition of the straw at sowing time, because the plantability of the subsequent crop can be hampered if performed soon after desiccation of the forages [51,52]. A large amount of straw on the soil surface can also increase the tractor’s slip during sowing, as well as causing “lodging” with accumulated straw between the lines of the seeder [53].

On the other hand, although the FPS of soybean after inoculated intercropping was lower, this treatment showed a higher GY (Table 2B), with an increase of 12% in grain yield (approximately 423 kg ha^−1^) (*p* < 0.05) compared to the non-inoculated treatment. This result can be justified both by the capacity of soybean to produce a greater number of branches to compensate for the lower plant stand and maintain productivity [54,55,56], as well as by the greater contribution of residues in this treatment, in this year (Figure 1), which justifies the higher grain yield of soybean in succession due to the benefits that the straw promotes within the system, such as soil protection that maintains moisture [57] and nutrient cycling [11,45,58]. In addition, the higher production of aerial biomass of cover crops suggests greater root growth [59], an important factor to improve the physical, chemical, and biological conditions of the soil [60,61] and increase the performance of the succession crop [11,62].

In 2016/17, there was an interaction between the cropping system and fertilization for GY (Table 2C). Soybean grown on sorghum monoculture residues with split fertilization (30–70%) had the highest GY compared to other fertilization managements, with 11.5% additional compared to GY obtained in sorghum monoculture residue fertilized with 100% N in cover, and 12.2% more than GY obtained in sorghum monoculture residue fertilized with 100% N in sowing.

The GY of soybean on monoculture sorghum residues with split fertilization was higher than soybean on sorghum monoculture residues fertilized with total N at sowing (Table 2C). In this case, the straw mineralization was probably influenced by N management, so that the splitting extended the availability of N over time and contributed to the mineralization of residues after the harvest of sorghum. This better synchronized the release of nutrients with the need of soybean in succession, while the total nitrogen fertilization only at sowing of sorghum may have kept the mineralization slower during the soybean cycle, due to the supply of N being at the beginning of the sorghum cycle and, therefore, much earlier than the sowing of soybean. This occurs because fertilization influences the communities and abundance of soil microorganisms, contributing to the mineralization of nutrients [63,64].

In the 2015/16 crop season, the insertion of first pod and the one hundred grain weight (HGW) of soybean on inoculated sorghum residues were higher (*p* < 0.05) compared to soybean on non-inoculated sorghum residues (Table 3). The greater contribution of residues in this treatment in this year (Figure 1) justified these results, due to the benefits that the straw promoted within the system, such as soil protection that maintains moisture [57], nutrient cycling [11,45,58], and improvement of the physical, chemical, and biological properties of the soils [60,61], improving the performance of the succession crop [11,62].

In 2016/17, the final stand of soybean plants on inoculated sorghum residues was higher compared to areas cultivated without inoculation, although the dry matter yield in the previous off-season did not differ (year 2016, Figure 1). This result can be attributed to the history of higher biomass yield and straw input in the 2015 off-season in this treatment (Figure 1), characterizing a residual effect. Under the same conditions, Nakao et al. [65] also observed the positive residual effect of inoculated grain sorghum with *A. brasilense*, intercropped with BRS Paiaguás grass, on the soybean in succession.

As observed for GY in the interaction between cropping modalities and *A. brasilense* in 2015/16 (Table 2B), the results for FPS (in the inoculation and cropping modalities) and plant height (PH) (in the cropping modalities) in 2016/17 (Table 3) were due to the various benefits of grass straw for the soil, such as maintaining moisture and controlling surface runoff, soil temperature, and nutrient cycling [11,45,57,58,66,67]. These factors condition the production environment to better support higher plant populations. The residual effect of cover crop management on soybean in succession was also reported by Kunrath et al. [62] as, according to them, the greater soil coverage with grass residues benefited the establishment of soybean and ensured a higher plant stand.

The highest FPS of soybean can also justify the highest PH, since the increase of interspecific competition causes the stem elongation of the plants [55]. This also provides higher insertion of the first pod (IFP); however, this effect did not occur in the present study (Table 3), which is advantageous, since higher IHFP results in plants with low stem exploration (that is, culms without pods) and, consequently, lower productive potential [55].

In 2016/17, the soybean GY was not influenced by the residual effect of the practices in the previous crops (Table 3), despite the differences in plant stands. As observed in 2015/16, this behavior was attributed to the compensatory capacity of the soybean crop to increase the number of branches to maintain the GY at similar levels to those obtained under the same condition, with different plant densities [54,55,56].

The nitrogen fertilizer management in the previous sorghum crop did not influence any of the evaluated variables (Table 3). Soybean was sown over the residues of the sorghum fertilized with the same dose of N in all treatments; only the fertilizer timing differed (0–100%; 30–70%; and 100–0% of the recommended amount of N at sowing and covering, respectively). Thus, the fertilizer management provided the same pattern of biomass yield in the grasses (sorghum and *Urochloa*) (Figure 1). This uniformity in the soil coverage between the treatments justified the absence of response of the soybean to the residual effect of the fertilizer in the previous crop.

Nitrogen is an ephemeral element in the soil; even with frequent fertilizations and relatively high doses [68,69], there is usually no direct residual effect on the crops grown in succession. However, the residual effect of nitrogen fertilization can occur indirectly, through the higher biomass yield of the fertilized cover crop, in which the greater input of dry matter to the soil will result in greater input and quantities of nutrients released to the crops in succession, by being fixed in organic compounds [11,50,70].

## 3. Materials and Methods

The experiment was conducted in the municipality of Selvíria, in the state of Mato Grosso do Sul, Brazil (20°20′05″ S and 51°24′26″ W, at 335 m above sea level), during two consecutive growing seasons: 2015–2016 and 2016–2017. Until 2013, cotton was grown on the experimental area, then the area remained fallow until the end of 2014. The climate in this region is Aw, characterized as humid tropical with a rainy season in summer and a dry winter, according to Koppen [5]. The long-term (1957–2014) average annual minimum and maximum temperatures were 18.4 and 31.3 °C, respectively. Monthly rainfall and the maximum and minimum temperatures during the experimental period were measured (Figure 2), as well as the photoperiod (Appendix A). The local soil was classified as a dystrophic Latossolo Vermelho with a clayey texture (580 g kg^−1^), according to the Brazilian System of Soil Classification [4] (Oxisol, USDA Soil Taxonomy).

Prior to the experiment installation, 20 soil samples were collected at 0.0–0.20 m depths and the soil chemical properties were determined (Table 4). Soil pH was measured in a 0.01 M CaCl_2_ suspension (1:2.5 soil/solution ratio). Organic C was determined with acidified Na_2_Cr_2_O_7_ to calculate soil organic matter (SOM).

Potential acidity (H  +  Al) was extracted with 0.5 M calcium acetate and determined by titration with 0.025 M NaOH solution. Exchangeable Al^3+^ was extracted with 1 M KCl at a soil:solution ratio of 1:10 and determined by titration with 0.025 M NaOH. Available P and exchangeable Ca^2+^, Mg^2+^, and K^+^ were extracted using ion exchange resin and determined using atomic absorption spectrophotometry. Cation exchange capacity (CEC) was calculated from the sum of Ca^2+^, Mg^2+^, K^+^, and H  +  Al. Base saturation (BS) was calculated by dividing the sum of Ca^2+^, Mg^2+^, and K^+^ by the CEC, and multiplying by 100%. All chemical analyses were performed following van Raij et al. [71].

There were two experiments with grain sorghum cropping. The experimental design was a randomized complete block design in a 2 × 2 × 3 factorial scheme, with four replications, consisting of sorghum monoculture cropped or intercropped with palisade grass, sorghum either inoculated or not with *Azospirillum brasilense*, and the application of nitrogen only at sowing or only in topdressing or split—30% at sowing and 70% in topdressing at the beginning of the panicle initiation stage—at 120 kg ha^−1^ N (Appendix A), using urea applied broadcast between the rows of sorghum. For all treatments, the basic fertilization in the sorghum sowing furrows consisted of 40 kg P ha^−1^ and 25 kg K ha^−1^.

The experiments were composed of 48 plots with seven rows of sorghum. The hybrid Ranchero with an aptitude for grain production was used. The diazotrophic bacteria were supplied by AZO Total Inoculant, developed for corn and wheat crops (registration number in MAPA: PR-93923-10074-1), physical nature: liquid, density: 1.0 g mL^−1^, use dosage: 100 mL^−1^ 20 kg seeds (guarantee of 2 × 10^8^ colony forming units mL^−1^ of *A. brasilense*, AbV5 and AbV6 strains). The sorghum seeds were manually inoculated by mixing them with inoculant in a plastic bag, about 30 min before sowing.

Sorghum was sown at a depth of 0.03 m using a no-till drill and a row spacing of 0.45 m. In the intercropping treatments, palisade grass cv. BRS Paiaguás was used in both years, and grass sowing was carried out simultaneously with the sorghum sowing and with another sowing-fertilizer machine with a double-disc type mechanism, disjoined for NT, and sown between sorghum lines. Grass seeds were sown at a 0.06 m depth, according to Kluthcouski et al. [72], with the objective of delaying the emergence of grass in relation to sorghum. It was not necessary to apply herbicide to suppress forage growth.

Nitrogen topdressing fertilization was hand-performed approximately 0.10 m from the sorghum plants, and was conducted between sorghum rows without incorporation according to the treatments (Appendix A), approximately 30 days after emergence (DAE) [73], in both growing seasons, when the plants were about 0.30 m high, when the plants had four expanded leaves (in April 2015 and May 2015), at the panicle initiation stage (growth stage 2) [15]. The experimental area was subsequently irrigated with 15 mm of water to minimize N losses due to volatilization.

After the sorghum harvest, the area was allowed to recover for 99 days until soybean planting. One day before grain sorghum harvest (in June 2015 and July 2016) and one day before desiccation for soybean sowing (in October of both years), the sorghum regrowth and grass were manually cut close to the ground. The dry matter production of the species was determined in two random samples per plot, using a 1 m^2^ metal frame. After the sampling, the area was desiccated with Glyphosate herbicide (1.44 kg ha^−1^ active ingredient [a.i.]) and after ten days, the area was managed by a horizontal plant-residue grinder, aiming for continuity of the no-tillage system. The soybean crop was implanted in succession both years and on the same sorghum plots.

Soybean was mechanically sown in November of each year, with the cultivar BMX Potência RR which is recommended for the region. A sowing-fertilizer machine was used with rod-type furrower mechanism (knife) for NT with a spacing of 0.45 m between rows and approximately 19 seeds per meter of furrow (Figure 3). Fertilization in the sowing furrow consisted of 26 kg ha^−1^ P and 50 K in the first year and 24 kg ha^−1^ P and 46 K in the second year (Appendix A).

Soybean seeds were treated with fipronil + pyraclostrobin (50 + 5 g a.i. 100 kg^−1^ seeds, respectively), and inoculated with *Bradyrhizobium japonicum* and *B. elkanii* (Semia 587 and Semia 5019 strains), using a mechanical mixer for incorporation. According to the needs of the crop, insecticides and fungicides were applied twice in both years (107.5 g ha^−1^ a.i. methomyl, 80 and 10 g ha^−1^ a.i. imidacloprid + beta-cyfluthrin and 60 + 24 g ha^−1^ a.i. azoxystrobin + cyproconazole + 0.5% mineral oil (*v*/*v*) or 107.5 g ha^−1^ a.i. methomyl, 10 g ha^−1^ a.i. chlorantraniliprole, 80 and 10 g ha^−1^ a.i. imidacloprid + betacyfluthrin and 66.5 + 25 g ha^−1^ a.i. of yraclostrobin + epoxiconazole).

The relative chlorophyll content (SPAD index—Soil Plant Analysis Development) and leaf nutrient concentrations were calculated during the flowering of the soybean crop at the full bloom (R2 growth stage) [74]; measurements were performed annually in January of each year. Digital chlorophyll meter (CFL 1030—Falker) was employed at the third totally developed trifoliate leaf, on the adaxial limbo side of median part, with an average of 10 readings per leaflet in ten plants per plot. Ten mature trifoliate leaves (3rd trifoliate) per plot were harvested [75] on the same day that SPAD readings were performed. Leaves were washed with deionized water and then dried under forced air circulation at 65 °C for 72 h before grinding and analyzing for chemical composition. Contents of N, P, K, Ca, Mg, and S were determined using methods described by [76].

For the harvest, in March of each year, the productive characteristics and grain yield of the soybean were evaluated. Plants were collected from each plot in the three central rows at 4 m (useful area), where the plant population was determined, extrapolated to 1 ha. In addition, ten plants were randomly collected per experimental unit and the insertion of first pod, total number of pods per plant, average number of grains per pod, and one hundred grain weight (moisture content: 130 g kg^−1^) were determined. In order to determine grain yield, 3 rows (2 m each) of the plot were harvested, mechanically threshed, weighed, and later calculated and extrapolated to kg ha^−1^ (moisture content: 130 g kg^−1^).

The data were submitted to the Shapiro–Wilk normality test [77] (*p* ≥ 0.05, *W* ≥ 0.90), and Levene’s test for homoscedasticity (*p* ≥ 0.05) [78], using the RStudio software, version 4.0.1 [79]; meeting the assumptions of statistical tests, ANOVA was performed. As recommended by [80], data were analyzed separately for each year due to the mean square values of extreme residuals (≥7). When significant, the means were compared using the Tukey test (*p* ≥ 0.05) using the SISVAR 5.3 software [81].

## 4. Conclusions

The inoculation of *A. brasilense* in sorghum for intercropping with palisade grass provided greater dry matter yield for soil cover regardless of nitrogen fertilizer management. Specific studies, however, are necessary to investigate the environmental influence on the efficiency of inoculation in sorghum, since the effect of this practice seemed to depend on the environmental conditions of each year.

The intercropping of sorghum with palisade grass was an efficient way of intensifying the production of biomass for animal grazing in a period historically characterized by food scarcity for herds, in addition to guaranteeing a greater amount of dry matter for the continuity of the no-till system.

The inoculation of sorghum by *A. brasilense* for intercropping with palisade grass resulted in higher soybean grain yield grown in succession (2015). This effect, however, varied between years, which suggests that the benefits of inoculation depend on seasonal factors that must be studied in future research.

## Figures and Tables

**Figure 1 plants-12-02007-f001:**
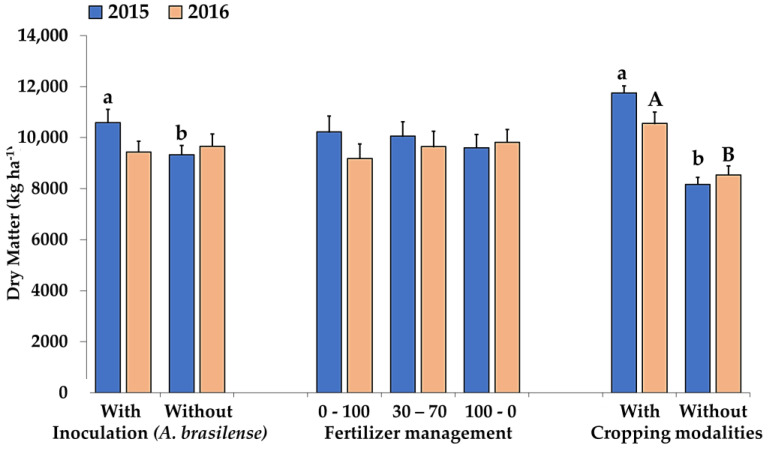
Aerial part dry matter accumulation for sorghum in two seasons. From left to right: either inoculated or not with *A. brasilense;* fertilized with 120 kg N ha^−1^ at different timings (0–100: N only at top-dressing; 30–70: split 30% at sowing and 70% at topdressing at the beginning of the panicle initiation stage; and 100–0: only at sowing); sorghum intercropped with grass or sorghum monoculture. Bars with different uppercase letters and lowercase letters denote a significant difference within each factor.

**Figure 2 plants-12-02007-f002:**
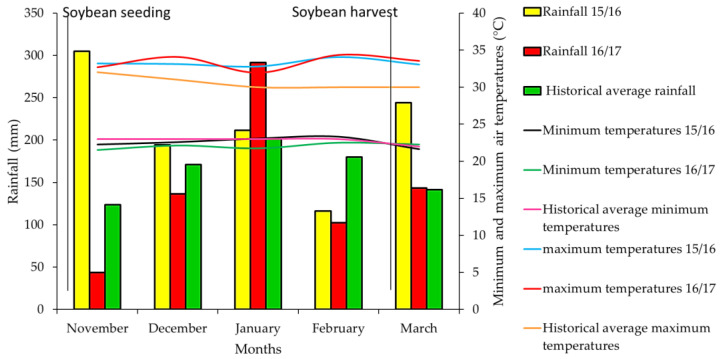
Accumulated rainfall (mm) and average minimum and maximum temperatures (°C) monthly during the field experiment period, and historical climate data (1980–2016) of the experimental site; Selvíria-MS, Brazil.

**Figure 3 plants-12-02007-f003:**
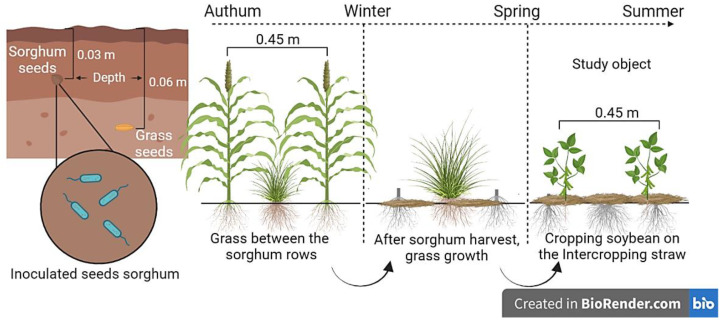
Schematic illustration of sorghum sowing in intercropping treatments with grass, and soybean cropping in succession, on intercropping residues.

**Table 1 plants-12-02007-t001:** SPAD index and average macronutrient values of soybean leaves at the full flowering phenological stage successive to grain sorghum under nitrogen fertilization management, with and without inoculation of *A. brasilense* in the seeds and grown alone or in intercropping with BRS Paiaguás grass for 2015/16 and 2016/17 ^(1)^.

**2015/16**							
**Treatments**	**SPAD**	**N**	**P**	**K**	**Ca**	**Mg**	**S**
	Units	-------------------------------------g kg^−1^-------------------------------------					
Inoculation							
With	49.1 ± 0.27	37 ± 0.82 b	3.5 ± 0.05	19 ± 0.54 b	6.0 ± 0.16 b	4.3 ± 0.11	2.1 ± 0.05
Without	50.0 ± 0.35	41 ± 0.93 a	3.6 ± 0.07	21 ± 0.48 a	7.0 ± 0.21 a	4.2 ± 0.04	2.1 ± 0.06
Cropping modalities							
Intercropped	49.8 ± 0.31 a	39 ± 1.22	3.5 ± 0.07	20 ± 0.52	6.4 ± 0.22 a	4.5 ± 0.09 a	2.1 ± 0.06
Monoculture	48.8 ± 0.29 b	39 ± 0.65	3.6 ± 0.05	20 ± 0.54	6.0 ± 0.16 b	4.1 ± 0.05 b	2.1 ± 0.06
Fertilizer management *							
0–100%	49.4 ± 0.38	39 ± 1.47	3.6 ± 0.10	20 ± 0.70 ab	6.1 ± 0.20 b	4.4 ± 0.13 a	2.2 ± 0.08
30–70%	49.2 ± 0.28	39 ± 1.35	3.5 ± 0.07	19 ± 0.71 b	6.8 ± 0.30 a	4.4 ± 0.09 a	2.2 ± 0.08
100–0%	49.4 ± 0.48	39 ± 0.61	3.6 ± 0.07	21 ± 0.38 a	6.0 ± 0.30 b	4.2 ± 0.06 b	2.0 ± 0.02
CV(%)	3	8	6	10	9	4	11
**2016/17**							
**Treatments**	**SPAD**	**N**	**P**	**K**	**Ca**	**Mg**	**S**
	Units	---------------------------------------g kg^−1^---------------------------------------					
Inoculation							
With	40.3 ± 0.46	46 ± 0.61 b	3.6 ± 0.04 b	13 ± 0.22 b	5.5 ± 0.12 b	3.5 ± 0.06	2.5 ± 0.03 a
Without	40.6 ± 0.57	49 ± 0.41 a	4.1 ± 0.09 a	15 ± 0.37 a	5.8 ± 0.09 a	3.4 ± 0.05	2.3 ± 0.05 b
Cropping modalities							
Intercropped	40.8 ± 0.43	46 ± 0.60 b	3.7 ± 0.06 b	13 ± 0.27 b	5.7 ± 0.10	3.5 ± 0.05 a	2.40 ± 0.03
Monoculture	40.3 ± 0.60	49 ± 0.44 a	4.0 ± 0.10 a	14 ± 0.39 a	5.6 ± 0.11	3.3 ± 0.06 b	2.3 ± 0.05
Fertilizer management *							
0–100%	40.4 ± 0.73	48 ± 0.61	4.0 ± 0.14	13 ± 0.42	5.6 ± 0.12	3.4 ± 0.08	2.4 ± 0.04
30–70%	40.9 ± 0.66	47 ± 0.77	3.7 ± 0.09	13 ± 0.51	5.9 ± 0.13	3.4 ± 0.07	2.3 ± 0.06
100–0%	40.1 ± 0.52	47 ± 0.74	3.9 ± 0.09	13 ± 0.30	5.6 ± 0.12	3.4 ± 0.07	2.3 ± 0.06
CV(%)	7	5	7	4	7	6	7

^(1)^ Means followed by different letter within each column are significantly different, Tukey’s (*p* ≤ 0.05). * Sowing and covering, respectively. Mean followed by standard error; SPAD: SPAD index.

**Table 2 plants-12-02007-t002:** Unfolding of the interaction between inoculation and cropping modalities for soybean plant stand in 2015/16 (**A**), grain yield in 2015/16 (**B**), and interaction between cropping modalities and nitrogen fertilizer management for soybean grain yield in 2016/17 (**C**), grown in succession to grain sorghum inoculated with *A. brasilense*, in monoculture or intercropped with BRS Paiaguás grass, and different nitrogen fertilizer management (120 kg N ha^−1^) ^(1)^.

	**Cropping Modalities**		
** *A. brasilense* **	**Intercropped**		**Monoculture**
(A)	Plants ha^−1^		
With	323,045 ± 6389 Bb		347,531 ± 5847 Aa
Without	344,444 ± 4260 Aa		337,654 ± 6925 Aa
(B)	Grain yield kg ha^−1^		
With	3887 ± 60 Aa		3946 ± 58 Aa
Without	3464 ± 93 Bb		3973 ± 74 Aa
	Nitrogen fertilizer management		
Cropping modalities	0–100% *	30–70%	100–0%
(C)	Grain yield, kg ha^−1^		
Intercropped	3686 ± 221	3494 ± 217	3808 ± 64
Monoculture	3587 ± 148 ab	4053 ± 134 a	3322 ± 128 b

^(1)^ Means followed by different uppercase letters within each column, and different lowercase letters in the line are significantly different, Tukey’s (*p* ≤ 0.05). * Seeding and coverage, respectively. Mean followed by standard error.

**Table 3 plants-12-02007-t003:** Soybean development and yield in succession to grain sorghum under management of nitrogen fertilization, with and without inoculation of *A. brasilense* in the seeds and monoculture grown or intercropped with BRS Paiaguás grass, 2015 and 2016 ^(1)^.

**2015/16**							
**Treatments**	**FPS**	**PH**	**FPI**	**NPP**	**NGP**	**HGW**	**GY ^(2)^**
Inoculation	(plants ha^−1^)	(m)	(cm)	-	-	(g)	(kg ha^−1^)
With	335,288 ± 4945	1.05 ± 0.02	18.4 ± 0.29 a	39 ± 0.83	2.3 ± 0.04	14.6 ± 0.22 a	3916 ± 041
Without	341,049 ± 4038	1.07 ± 0.01	16.9 ± 0.28 b	41 ± 0.93	2.3 ± 0.08	13.8 ± 0.27 b	3719 ± 78
Cropping modalities							
Intercropped	333,745 ± 4368	1.06 ± 0.01	17.3 ± 0.31	40 ± 0.85	2.3 ± 0.06	14.0 ± 0.19	3775 ± 69
Monoculture	342,593 ± 4550	1.05 ± 0.02	17.9 ± 0.32	40 ± 0.96	2.2 ± 0.06	14.3 ± 0.32	3959 ± 46
Fertilizer management *							
0–100%	343,519 ± 6077	1.07 ± 0.01	17.5 ± 0.35	40 ± 1.38	2.2 ± 0.09	14.0 ± 0.32	3769 ± 82 ab
30–70%	334,259 ± 5625	1.05 ± 0.02	17.9 ± 0.46	40 ± 0.76	2.3 ± 0.06	14.0 ± 0.35	3728 ± 83 b
100–0%	336,728 ± 4876	1.06 ± 0.02	17.5 ± 0.37	41 ± 1.13	2.3 ± 0.08	15.0 ± 0.28	3955 ± 66 a
CV(%)	6	6	8	11	14	9	6
**2016/17**							
**Treatments**	**FPS**	**PH**	**FPI**	**NPP**	**NGP**	**HGW**	**GY ^(3)^**
Inoculation	(plants ha^−1^)	(m)	(cm)	-	-	(g)	(kg ha^−1^)
With	286,806 ± 5857 a	1.05 ± 0.71	16.8 ± 0.28	39 ± 1.04	2.6 ± 0.06	15.0 ± 0.53	3749 ± 114
Without	267,130 ± 6662 b	1.05 ± 1.13	16.2 ± 0.18	41 ± 0.84	2.5 ± 0.06	14.2 ± 0.32	3568 ± 95
Cropping modalities							
Intercropped	287,963 ± 6315 a	1.07 ± 0.73 a	16.1 ± 0.24 b	39 ± 1.09	2.5 ± 0.05	14.5 ± 0.47	3663 ± 110
Monoculture	265,972 ± 6059 b	1.03 ± 1.01 b	16.8 ± 0.22 a	41 ± 0.78	2.5 ± 0.07	14.7 ± 0.41	3654 ± 103
Fertilizer management *							
0–100%	274,306 ± 7098	1.05 ± 1.33	16.5 ± 0.30	40 ± 0.87	2.6 ± 0.10	14.3 ± 0.44	3636 ± 129
30–70%	279,861 ± 8664	1.07 ± 1.03	16.6 ± 0.32	40 ± 0.75	2.5 ± 0.05	14.9 ± 0.61	3724 ± 156
100–0%	279,861 ± 8595	1.04 ±1.02	16.4 ± 0.29	40 ± 1.70	2.5 ± 0.07	14.6 ± 0.56	3615 ± 103
CV(%)	11	4	7	13	12	16	12

^(1)^ Means followed by different letters within each column are significantly different, Tukey’s (*p* ≤ 0.05). Mean followed by standard error. ^(2)^ Interaction between the inoculation and cropping modalities. ^(3)^ Interaction between cropping modalities and nitrogen fertilizer management. * Sowing and covering, respectively. SMD: significant minimum difference; FPS: final plant stand; PH: plant height; IFP: insertion of first pod; NPP: number of pods per plant; NGP: number of grains per pod; HGW: one hundred grain weight; GY = grain yield.

**Table 4 plants-12-02007-t004:** Soil chemical characteristics in the experimental area before the onset of the experiment.

Depth	pH	SOM ^a^	P	H + Al	K^+^	Ca^2+^	Mg^2+^	CEC ^b^	BS ^c^
m	CaCl_2_	g dm^−3^	mg dm^−3^	---------------mmol_c_ dm^−3^ ---------------					%
0–0.20	5.5	22	17	28	1.4	26	18	73.1	62

^a^ SOM, soil organic matter; ^b^ CEC, cation exchange capacity at pH 7.0; ^c^ BS, base saturation.

## Data Availability

Not applicable.

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
