# Peer review of "Soybean Yield and Nutrition Grown on the Straw of Grain Sorghum Inoculated with *Azospirillum brasilense* and Intercropped with BRS Paiaguás Grass"

_plants, 2023, doi:10.3390/plants12102007_

Round 1
Reviewer 1 Report
Dear Authors,
manuscript Soybean yield and nutrition grown on the straw of grain sorghum inoculated with Azospirillum brasilense and intercropped with BRS Paiaguás grass which I had the pleasure of reviewing in my opinion is very well written and provides very relevant information and guidance for agronomy.
Regarding the content of the manuscript, I have a few suggestions:
Keywords: I believe that the Latin name of maize instead of sorghum was given incorrectly.
Introduction: I suggest adding information about grain and forage quality in sorghum. I also suggest mentioning the relevance of soybean crops and their importance to the economy.
Materials and methods: a table from the supplementary materials on soil is suggested in the manuscript near lines 275 - 286
Author Response
Reviewer 1
Dear Reviewer,
Thank you for all comments and suggestions made in our manuscript, certainly it improved its quality.
Please find our answers to your points below.
Note: Modifications to the article were indicated in yellow in the attached manuscript.
- Keywords: I believe that the Latin name of maize instead of sorghum was given incorrectly.
ANSWER: That’s right, the latin name of maize instead of sorghum was given incorrectly (line 32).
- Introduction: I suggest adding information about grain and forage quality in sorghum. I also suggest mentioning the relevance of soybean crops and their importance to the economy.
ANSWER: As suggested, were included in the introduction information about sorghum and the importance of soybean crop to the economy (lines 68-70 and 79-85), with more details for soybean, which is the focus of the work
- Materials and methods: a table from the supplementary materials on soil is suggested in the manuscript near lines 275 – 286.
ANSWER: The table was inserted as suggested (line 292).
Reviewer 2 Report
Dear Colleagues.
Dear colleagues, There are several questions. They are in the attached file.
Sincerely yours

Author Response
Dear Reviewer,
Thank you for your positive and helpful comments, certainly it improved our manuscript.
Please see the attachment.
Note: Modifications to the article were indicated in yellow in the attached manuscript.

Reviewer 3 Report
This paper investigated the dry matter accumulation of sorghum with or with-out Azospirillum brasilense inoculation, nitrogen fertilization split and intercropped with palisade grass, and how these practices influenced the nutrition and development of soybean in succession. Following are the comments:
1.Difference between the two growing seasons (2015 and 2016) should be showed in results and discussed.
2.The language should be edited more carefully, for example variable symbol should be in italic.
3. Line 109: To benefit the readers, it is better to explain X-axis in Figure 1 from left to right.
4.Line231: pH
5.Line317: check the sentence.
Author Response
Reviewer 3
Dear Reviewer,
We would like to sincerely thank you for your time spent reviewing our manuscript, the comments and suggestions which significantly improved our manuscript.
Please find our answers to your points below.
Note: Modifications to the article were indicated in yellow in the attached manuscript.
- Difference between the two growing seasons (2015 and 2016) should be showed in results and discussed.
ANSWER: We had the objective of evaluating the difference between the two growing seasons (joint statistical analysis of data from 2015 and 2016), but as recommended by Pimentel-Gomes (2000), the data were analyzed separately for each year, given the het-erogeneity between the mean squares of extreme residuals (values greater or equal to 7). This was probably due to the peculiar climatic conditions in each year.
- The language should be edited more carefully, for example variable symbol should be in italic.
ANSWER: It was corrected in the text.
- Line 109: To benefit the readers, it is better to explain X-axis in Figure 1 from left to right.
ANSWER: Corrected it as suggested (lines 118-123).
- Line 231: pH
ANSWER: As mentioned in the previous paragraph (line 236) “PH” means plant height.
- Line 317: check the sentence.
ANSWER: It was corrected in the text (332-333).